# Associations of Motor Performance and Executive Functions: Comparing Children with Down Syndrome to Chronological and Mental Age-Matched Controls

**DOI:** 10.3390/children9010073

**Published:** 2022-01-05

**Authors:** Thomas Jürgen Klotzbier, Benjamin Holfelder, Nadja Schott

**Affiliations:** Department of Sport and Exercise Science, Institute of Sport Psychology & Human Movement Performance, University of Stuttgart, 70569 Stuttgart, Germany; benjamin.holfelder@inspo.uni-stuttgart.de (B.H.); nadja.schott@inspo.uni-stuttgart.de (N.S.)

**Keywords:** intellectual disability, executive function, modified trail-making test, movement assessment battery, mental age paradigm

## Abstract

Background. Children with Down syndrome (DS) exhibit lower motor and cognitive performance than typically developing children (TD). Although there is a relationship between these two developmental domains, only a few studies have addressed this association in children with DS compared to groups of the same chronological age (CA) or mental age (MA) within one study. This study aimed to fill this research gap. Method and Procedures. The Movement Assessment Battery for Children-2 and the Trail-Making Test was used to assess motor and cognitive performances in 12 children (M = 10.5 ± 10.08) with DS, 12 CA-matched, and 12 MA-matched controls. Results. There are significant group differences in the motor dimension (total test score; *p* < 0.001, η^2^_p_ = 0.734), for processing speed (*p* < 0.001, η^2^_p_ = 0.396), and cognitive flexibility (*p* < 0.001, η^2^_p_ = 0.498). Between TD-CA and both other groups, the differences in the magnitude of correlations for the motor dimension balance are also significant (compared to DS: z = −2.489; *p* = 0.006, and to TD-MA: z = −3.12; *p* < 0.001). Conclusions. Our results suggest that the relationships depend on the studied cognitive and motor skills. It seems crucial to select a wide range of tasks for both domains that are as isolated as possible for future studies, to better understand the relationships between cognitive and motor skills in children with DS.

## 1. Introduction

Down syndrome (DS) is a genetic neurodevelopmental disorder associated with delayed motor and mental development [1]. The developmental trajectories of children with DS are highly variable. One of the most evident features of DS is the impairment in cognitive development [2] and weaknesses in motor development and control [3,4].

Regarding cognition, the main areas affected are language skills, processing speed, attention processes, visuo-spatial abilities, and, specifically, a reduced ability in executive functions (EF) [5,6]. The construct of EF has received considerable attention over the last three decades. Despite, or perhaps because of, the many publications, there is no generally accepted definition of EF [7]. Frequently, EF are divided into three domains, cognitive flexibility, inhibition ability, and working memory, and encompass an extensive set of higher-order operations that organize and regulate goal-directed behavior within the prefrontal cortex (PFC) [8]. In their meta-analysis, Tungate and Conners [5] were able to show that there is a clinically significant overall weakness in EF for individuals DS with a relative strength in inhibition ability relative to TD children with the same mental age.

In addition to reduced cognitive performance, motor performance is also impaired [3]. Most notable is the reduced speed of movement execution and decreased precision of a multitude of skills, with object manipulation skills [9] and postural control [10] particularly affected. Uncoordinated, slower, more variable, and hesitant movements, along with a poor ability to respond to environmental changes, are characteristics of motor skill performance in children with DS [11]. Additionally, motor abilities such as coordination, balance, and strength [12] may not be as developed as the those of peers without DS would show. Using various approaches, further studies showed that children with DS between the ages of 6 and 16 years performed worse on fundamental movement skills than their typically developing peers [13,14,15]. Volman et al. [14] observed that children with DS scored poorly on manual dexterity, followed by balance and then ball skills with high interindividual variability.

The cross-domain effects of impairments in EF play a decisive role in the motor control deficits described. Although there is general agreement that motor and cognitive development are closely linked and have similarly protracted developmental trajectories [16], the extent of the interaction between the cognitive profile and motor control in individuals with DS is largely unexplored, with positive, albeit small to moderate, correlations [17,18]. Westendorp et al. [19] provide several approaches to explain this relationship, such as the cerebellum’s role, a similar developmental timetable with an accelerated development for both domains between 5 and 10 years of age, and several common underlying processes such as sequencing, monitoring, and planning. Wassenberg et al. [20] discovered a positive but small relationship between motor performance and general cognitive performance in a sample of 5- to 6-year-old typically and atypically performing children. According to Hartman et al. [21], intellectually disabled children are, in addition to the impairments in qualitative motor skills, also impaired in higher-order cognitive functions (e.g., EF). The authors state that the deficits in the two domains are interrelated and inextricably intertwined.

The different—mainly correlative—studies on the relationship between the two domains use various measurement methods to assess motor skills and cognitive abilities [22]. Due to the small number of studies on individuals with DS and the different cognitive tasks used, no clear conclusions can be drawn about the relationship between motor skill performance and EF. Schott and Holfelder [18] published a study examining the relationship between motor skill performance and EF in children with DS. The authors showed that motor skill performance (using the TGMD-3, [23]) and EF performance (using the Trails-P, the Trail-Making Test (TMT) for young children, adapted from [24]) are positively correlated in children with DS and that children with DS have significant deficits in motor tasks in addition to impairments in cognitive functions. Specifically, they found that locomotor skills and task D (distraction) of the Trails-P were highly correlated (r = 0.80) in children with DS. Furthermore, significant correlations were found between object control skills and task A (baseline control, r = 0.54), task B (attentional control, r = 0.61), and task D (r = 0.60). For typically developed (TD) children, analyses revealed no significant correlations between Trails-P and locomotor or object control skills. A limitation of this study is that no control group of the same mental age (MA) was included. Although the study’s sample size of Schott and Holfelder [18] is similar to comparable studies [25,26], a statement on generalizability was only possible to a limited extent. In addition, the cognitive demands in the study mentioned above may have been too low for TD children; a ceiling effect of cognitive performance was observed in this group. 

Therefore, a similar study design with related research objectives, supplemented with subjects with the same mental age and equivalent methods (in terms of a rough replication), would increase the significance and lead to a more general and more meaningful scope of interpretation. A comparison with an MA group is appropriate to determine whether the reduced performance in children with DS is “only” a developmental delay. If children with DS are disadvantaged compared to children of the same MA, the “Conventional Difference Hypothesis” postulated by Milgram [27] is considered valid. According to this hypothesis, children with DS will always prove inferior because intelligent quotient (IQ) and not mental age predicts problem-solving ability [28]. If, on the other hand, children with DS show better performance than TD-MA children due to their CA progression and greater experience, an “Unconventional Differential Hypothesis” postulated by Kohlberg [29] is considered appropriate (see also [30]). Kohlberg argued that children with intellectual disabilities are richer in “general experience” than younger children of the same MA and claimed that this additional experience provides a performance advantage. 

Thus, the present study examines the relationships between EF and motor skill performance in children with DS and typically developing children of the same mental (TD-MA) or chronological age (TD-CA). Based on the described motor and cognitive impairments in children with DS and according to the “Conventional Difference Hypothesis”, we predict that typically developing children in both control groups (TD-CA and TD-MA) will perform better in motor skills performance and EF compared to children with DS. Furthermore, we assume that the relationship between the EF and motor skill performance will become stronger with increasing cognitive task demands and that the relationship will be stronger in children with DS compared to both control groups [31]. In particular, the relationship should be most robust in tasks with a high cognitive load. 

## 2. Methods

### 2.1. Participants

Thirty-six Caucasian children (*n* = 18 female; 8.61 ± 2.52 (range 4–11) years) were recruited from the Rhein-Neckar region (Germany). TD children were recruited from schools and kindergartens, while the children with DS were recruited from a school for children with special educational needs. Based on an investigator’s email request, the children’s legal guardians volunteered and agreed to their child’s participation in the study. The Peabody Picture Vocabulary Test—IV (PPVT-IV; Dunn & Dunn [32]; German adaptation by Lenhard et al. [33]; for detailed information, see Section 2.2.3.2) was used to assess the participants’ MA. Based on the PPVT-IV results, TD children were assigned to the control groups. The inclusion criteria of the twelve children with DS are (a) age 8–12 years (to compare the results by Schott and Holfelder [18] and due to the fact that children with DS are only able to perform the TMT at this age range), (b) physician-diagnosed DS, (c) ability to follow simple instructions, (d) ability to walk independently, (e) proficiency in numbers up to 25, (f) knowledge of the letters of the alphabet, and (g) normal/corrected vision. Children with comorbidities such as autism spectrum disorders, cerebral palsy, deafness, blindness, or other neuro-musculoskeletal disorders were excluded from the present study. The TD children were free of developmental delays or physiological impairments. All the inclusion criteria mentioned were verified by asking the parents and the educators.

#### Matching Procedure

A TD child was included in the TD-MA control group if their raw score in the PPVT-IV was less than four standard deviation points (within the four SD range of children with DS) away from the corresponding mean score of children with DS. A TD child was included in the TD-CA group if their CA was within the 4-month range of the children with DS. 

### 2.2. Materials

#### 2.2.1. Motor Performance

The Movement Assessment Battery for Children-2 (MABC-2; [34]) was used to assess the three motor development dimensions of manual dexterity, ball skills, and the ability to perform static and dynamic postural control. The children were asked to perform three activities in the manual dexterity subtest (e.g., placing pegs, threading a lace, drawing trail), two activities in the aiming and catching category (e.g., catching with two hands, throwing a beanbag on to a mat), and three activities in the balance category (e.g., one-board balance, walking heel-to-toe forward, hopping on a mat) according to their respective age band (AB; AB1: 3 to 6 years; AB2: 7 to 10 years; AB3: 11 to 16 years). For the motor dimension percentiles, each task’s raw score in the MABC-2 was converted to a standard score, and a total test score (TTS) was calculated by summing the eight task standard scores. Using the standard score and the TTS, a percentile score can be obtained from the norm tables published in the MABC-2 manual [35] to screen for a child’s motor delays or disorders. The percentile scores are described as a traffic light scoring system, including a red, an amber, and a green zone. Values at and below the 5th percentile indicate significant motor deficits (red zone). Children who achieve a test value between the 6th and 15th percentile are classified in the high-risk group (amber zone). Values that exceed a percentile rank of 15 are considered inconspicuous (green zone). The use of percentiles allows a direct comparison of correlations considering the differences in mental and chronological age. The test reliability after two weeks is r = 0.97 (N = 138; [34]). According to Blank et al. [36], the MABC-2 show good-to-excellent interrater reliability, good-to-excellent test–retest reliability, and fair-to-good validity.

#### 2.2.2. Cognitive Performance

The Trail-Making Test (TMT; [37]) was used as a standardized neuropsychological test to assess EF under fine motor control conditions. In its original version, the paper-and-pencil test consists of two parts. In Part A (TMT-A), participants are instructed to connect numbers (1–25) in ascending order. This condition is less demanding and requires in particular information processing speed. In Part B (TMT-B), the participants are instructed to connect randomly positioned numbers (1–13) and letters (A-L) in an alternating ascending sequence (e.g., 1-A-2-B-3-C….). This condition is more demanding and places greater load on EF, especially cognitive flexibility [38,39]. We also included a motor speed condition (TMT motor speed; TMT-M). In this condition, the participants follow a given path of equal length as in the TMT A [40,41]). The time was measured with a stopwatch to the nearest 00.01 s. To account for the different lengths in the paper–pencil version of the TMT (TMT-M = 185.4 cm; TMT-A = 185.4 cm; TMT-B = 243.8 cm; [42]) when reporting the times required, the velocities of each condition were first calculated and then standardized to a length of 200 cm: Times in TMT (𝑠) = 200/velocity in TMT (time needed for 200 cm). The normalized time for the pure motor condition (TMT-M) was subtracted from the normalized time for the TMT-A condition to calculate the “pure” cognitive information processing: Information processing = TMT-A200 − TMT-M200. The normalized time for the “pure” motor condition (TMT-M) was subtracted from the normalized time for the TMT-B condition and subtracted by the purely cognitive information processing speed in order to calculate pure cognitive flexibility: Cognitive flexibility = (TMT-B200 - TMT-M200) − (TMT-A200 − TMT-M200). A value of r = 0.94 (TMT-A) to 0.90 (TMT-B) was calculated for the test reliability of the TMT [43]. For background, psychometric properties, administration procedures, and interpretive guidelines of the TMT, we refer to [44,45].

In addition to cognitive processing speed, linguistic, executive, and attentional components are also assessed [46]. Thus, it is difficult to distinguish between the different cognitive components that are required to complete the TMT. Various components of EF play a role in processing the TMT. For example, the TMT provides information on performance in visual search, information processing, fine motor skills, cognitive flexibility, and other EF [47]. It is probably the most widely used instrument for assessing task switching ability [39,48], and Part B of the TMT is also often referred to as the “frontal lobe test” [49,50], which is strongly associated with EF.

#### 2.2.3. Covariates

##### 2.2.3.1. Sociodemographic Information and Sports Participation 

Sociodemographic and health characteristics included age, sex, body composition, medication, and sports participation (three activities, duration, and frequency). Children’s height (m), weight (kg), palm length, and middle finger length (cm) were measured, and the body mass index (BMI, kg/m^2^), as well as the palm-to-finger-length ratio, were calculated. The parents were also asked how many days per week and minutes per session their children had participated in each activity. The total sports participation (h/week) was then calculated as follows: (frequency_activity1 × duration_activity1) + (frequency_activity2 × duration_activity2) + (frequency_activity3 × duration_activity3) [41]. 

##### 2.2.3.2. Receptive Vocabulary Test

The Peabody Picture Vocabulary Test (PPVT-IV [32]; German adaptation [33]) was used as a measure of the mental age (MA), an assessment for measuring verbal skills in the standard American English vocabulary (here German vocabulary). It can be utilized to measure the receptive processing of vocabulary in individuals with ID [51]. Krasileva and colleagues [52] used the PPVT-IV scores as proxy for IQ in studies of autism spectrum disorder. The test is available as a paper version and contains 228 items, consisting of a spoken word and an associated set of four colored pictures. The subject’s task is to select the picture that best matches the test administrator’s word. The 228 items of the PPVT-IV are grouped into 19 item sets of 12 items each. The sets are arranged in ascending order of difficulty so that only those sets can be applied that are appropriate for a child’s particular level of difficulty (performance range). Depending on the age of the TD children, an item set is selected as the entry point. For children with DS, the lowest difficulty level of the test was used initially because of the high inter-individual variability in the degree of intelligence impairment (ranging from IQ values of 20–69; [53]). Test–retest reliability is r = 0.91 for the German sample for a period of 6 to 12 months. The measure of internal consistency across all study blocks is reported with a Cronbach’s alpha of 0.87. Additionally, a split-half correlation of 0.97 (N = 4532) is reported across all ages [33]. 

### 2.3. Experimental Procedure

Upon arrival at the Cognitive and Motor Research Laboratory on the first day, parents signed the informed consent form and completed a questionnaire on their child’s sociodemographic data and sports participation. Subsequently, the TMT and then the MABC-2 were administered. To avoid cognitive and physical fatigue from collecting data through the MABC-2 and to avoid loss of attention that could influence performance, the cognitively demanding TMT was conducted for all participants prior to the MABC-2. Children with DS comprehend and apply visual–motor instructions better than verbal ones [54,55] and have a remarkable ability in imitation processes [56]. For these reasons, the tasks were explained verbally, and a practical demonstration was provided. For each of the three conditions of the TMT, there was a short trial version. On the second day, children completed the PPVT-IV (to divide the groups into MA and CA), and the body composition measurements were taken. The test duration was about 60 min on both days. Sufficient breaks were given between tests to avoid physical overload and maintain optimal cognitive and physical performance for all children. All testing rooms were bright and quiet, and there was a table and chairs in the rooms so that tasks could be performed while sitting to achieve the greatest possible standardization. All assessments were conducted in accordance with ethical rules for research in human subjects following the Declaration of Helsinki [57]. The studies involving human participants were reviewed and approved by the University of Stuttgart. The children were asked for their consent and their willingness to participate in the study and the participant’s legal guardian/next of kin provided written informed consent to participate in this study. The participants or the legal guardians of the children did not receive any financial compensation or incentive for taking part in the study.

### 2.4. Data Analysis

All statistical analyses were carried out with SPSS v.27 (SPSS, Chicago, IL, USA). The Kolmogorov–Smirnov test was used to test each variable for normal distribution. For sample characteristics, possible group differences for continuous variables (e.g., age, height, weight, BMI, sports participation), were calculated using *t*-tests. Categorial demographic variables (e.g., sex) were tested with a Chi^2^ test. If there were significant results from the (M)ANOVAs, post-hoc tests (Bonferroni correction) were used to determine which factor levels differed significantly (*p* values set to 0.05; [58]). Effect sizes for all ANOVAs were reported using partial eta-squared (η^2^_p_). There were no missing data. A MANOVA was calculated with group as a fixed factor and percentiles of the motor dimensions as dependent variables to show group differences in the motor dimensions. A 3 (group: DS; TD-MA & TD-CA) × 2 (information processing and cognitive flexibility) ANOVA with repeated measurements was performed to test the different cognitive performance effects. A MANOVA was performed with information processing and cognitive flexibility as fixed factors to test the group differences in cognitive performance.

Partial correlations controlling for sports participation and sex were calculated separately for children with DS and TD children (TD-MA, TD-CA) to measure associations between cognitive (TMT) and motor (MABC-2) indices. A logarithmic transformation (base 10) was applied to each participant’s speed scores in the TMT conditions to obtain a normal distribution for the cognitive indices. For the motor indices, the percentiles of the motor dimensions of the MABC-2 were used. Correlations were deemed significant if *p* < 0.05. Fisher’s z-score transformations and *t*-tests were applied using freeware [59] to determine whether DS and TD-MA or TD-CA children showed different correlations. 

## 3. Results

### 3.1. Participants

The mean age of children with DS is 10.5 ± 10.08, TD-CA children 10.5 ± 10.07, and TD-MA children 5.98 ± 1.21. All groups have a sex distribution of 50%. None of the children were obese or overweight; Body Mass Index (BMI) did not differ significantly between children with DS and TD-CA children (see Table 1). On average, children with DS exercise 138 min per week (SD = 45.1), comparable to TD-CA children’s values but significantly different from TD-MA children. The characteristics for BMI, sports participation, and motor performance are comparable to recently published data for children of the same age with DS (see [60] for BMI, [61] for sports participation and [62] for motor coordination). 

The children with DS have a low raw score in PPVT-IV (M = 96.6, SD = 19.7), which differs significantly from the TD-CA group’s scores and has a large effect size. This corresponds to the diagnostic criteria and the international classification of mental disorders [53]. 

Overall, only moderate correlations exist between demographic characteristics, body composition, and motor performance (see Appendix A). 

### 3.2. Motor Performance

Figure 1 shows the groups’ mean percentiles on all dimensions and the TTS score of the MABC-2. The MANOVA shows significant group differences for the manual dexterity percentiles, F(2,33) = 35.6, *p* < 0.001, η^2^_p_ = 0.683. The pairwise comparisons show that children with DS differ significantly from TD-MA and TD-CA (*p* < 0.001), but TD-MA does not differ from TD-CA (*p* = 0.463). There is a significant group difference for aiming and catching percentiles, F(2,33) = 190.0, *p* < 0.001, η^2^_p_ = 0.535, with all groups differing significantly (DS vs. TD-MA, *p* < 0.002; DS vs. TD-CA, *p* < 0.001; TD-MA vs. TD-CA, *p* = 007). There is also a significant group difference for the percentiles of the motor dimension B, F(2,33) = 28.7, *p* < 0.001, η^2^_p_ = 0.635. The multiple comparisons again show that all groups differ from each other (DS vs. TD-MA, *p* < 0.001; DS vs. TD-CA, *p* < 0.001; TD-MA vs. TD-CA, *p* = 030). Additionally, with regard to the TTS, there are significant group differences, F(2,33) = 45.6, *p* < 0.001, η^2^_p_ = 0.734, with all groups differinf significantly (*p* < 0.001) (see Figure 2). The CA matched group are performing better than might be expected as their percentile scores are all at or above 50%. 

### 3.3. Cognitive Performance

The absolute times in the TMT conditions show significant (*p* < 0.001) group differences for all conditions, with the lowest times observed in the TMT-M (DS: M = 89.7, SD = 33.8; TD-MA: M = 54.6, SD = 23.4; TD-CA: M = 32.3, SD = 11.9), compared with TMT-A (DS: M = 146, SD = TD-MA: M = 178, SD = 81.4; TD-CA: M = 32.7, SD = 50.01) or TMT-B (DS: M = 349, SD = 98.2; TD-MA: M = 264, SD = 91.9; TD-CA: M = 80.2, SD = 20.7). There is no relationship between finger-to-palm ratio and performance in the pure motor task (TMT-M), r(36) = 0.100, *p* = 0.569), which could have indicate an influence on hand motor function. A 3 (group: DS, TD-CA & TD-MA) × 2 (cognitive function: information processing and cognitive flexibility) ANOVA with repeated measurement for normalized times in the TMT show significant main effects for group, F(2,33) = 18.6, *p* < 0.001, η^2^_p_ = 0.530, and a significant interaction group × cognitive function, F(2,33) = 10.1, *p* < 0.001, η^2^_p_ = 0.380. The interaction illustrates that children with DS have difficulties primarily with higher cognitive demands.

The MANOVA to test the group differences in cognitive function (information processing and cognitive flexibility) shows significant group effect in information processing, F(2,33) = 10.8, *p* < 0.001, η^2^_p_
*=* 0.396, and in cognitive flexibility, F(2,33) = 16.3, *p* < 0.001, η^2^_p_
*=* 0.498, with higher differences between groups for cognitive flexibility. Post-hoc analysis for information processing show that all three groups differed from each other (DS vs. TD-MA: *p* = 0.016; DS vs. TD-CA: *p* = 0.043; TD-MA vs. TD-CA: *p* < 0.001), with TD-MA children in particular producing the lowest performance in information processing (M = 134, SD = 77.4). Smaller differences between TMT-M and TMT-A (M = 0.442, SD = 11.6) in TD-CA children indicates better information processing. DS and TD-MA (*p* < 0.001) and DS and TD-CA (*p* > 0.001) differ from each other on cognitive flexibility, with children with DS exhibiting the lowest performance in cognitive flexibility (M = 128, SD = 65.5). However, differences between TD-MA (M = 24.1, SD = 54.8) and TD-CA (M = 30.5, SD = 17.6) are not observed (*p* = 0.760) (see Figure 3).

### 3.4. Relationship between Motor Skill and Cognitive Performance

Table 2 reports the partial correlations controlled for the children’s sports participation and sex between the cognitive (TMT) and motor (MABC-2) indices. Correlations with medium-to-high effect sizes (convention according to Cohen [63]) are obtained between all TMT conditions and the dimensions manual dexterity, aiming and catching, balance, and the Total Test Score of the MABC-2 in all groups. In the group with children with DS, lower correlations are found for almost all relationships. Additionally, there are almost no significant differences in the magnitude of correlations that emerge between the TMT and the MABC-2. The only significant correlation can be observed in TD-CA children for the TMT with high cognitive load and the percentiles in the motor dimension balance and TTS. The differences in the magnitude of the motor dimension balance correlations between TD-CA and both other groups are also significant (see Table 3), with TD-MA and children with DS showing negative correlations and TD-CA children showing positive correlations (see Table 2). Differences in the magnitude of the correlations between DS and TD-MA children can only be observed for the motor dimension aiming and catching, with children with DS showing negative correlations and TD-MA children showing positive correlations.

## 4. Discussion

The study aimed to compare motor skill performance (MABC-2) and cognitive performance (TMT) and the interaction of both domains in children with DS and TD children (matched for CA and MA). As expected, children with DS showed lower performance in all motor dimensions and reduced cognitive performance compared to TD-MA and TD-CA children. Regarding cognitive performance, DS and TD-MA children’s differences depended on the cognitive domain, with children with DS having severe difficulties with cognitive flexibility tasks. Concerning the associations between motor and cognitive domains, an association can be observed mainly in TD-CA children. In TD-MA children and children with DS, we only saw a small non-significant correlation between selected tasks. 

### 4.1. Motor Skill Performance 

Regarding the motor performance measured with the MABC-2, TD-CA children achieved the best results in all three motor dimensions, as expected, with significant differences between the TD-MA and TD-CA groups only for the dimension aiming and catching and the TTS. Like the studies of Gardner and Broman [64] and Mathiowetz et al. [65], manual dexterity performance in individuals with DS lagged behind their CA-matched and MA-matched peers. Considering the significant differences in age and sports participation between TD-MA and TD-CA children, significant differences were expected for the other two sub-dimensions aiming and catching and B. This is because motor performance is highly experience-dependent [66] and is associated with increasing age and, in particular, organized sport [67], which was not captured within the present study. In addition, all subscores of the TD-MA and TD-CA children are significantly above the 16th percentile, indicating an age-expected norm [34], which is not the case for children with DS, especially not for manual dexterity and B. These results are consistent with previous studies in children with DS as a consequence of difficulties in balance, postural control, as well as fine and gross motor tasks [67] (e.g., [3,18,40]). 

### 4.2. Cognitive Performance

Regarding the performance of EF, TD-CA children achieved the best results for all three TMT conditions, as expected. These results are also confirmed by comparing data from young adults from different countries [68]. However, exciting findings emerge when comparing the results of the children with DS with those of the TD-MA. While the children with DS have significantly more difficulties with the TMT motor speed (fine motor performance) than the TD-MA group, it also reflects the results of the MABC-2 sub-dimension manual dexterity. In contrast, the children with DS achieved significantly better results on the TMT-A (information processing speed) compared to TD-MA. This observation is surprising because individuals with DS are impaired in information processing [69] and cognitive functions such as EF [5].

On the other hand, these findings represent EF’s nature, as EF’s development is strongly dependent on experience [70] and brain maturation [71]. In Part B, both TD-MA children and children with DS have significantly greater difficulties with the task. This is not surprising since the TMT-B places high demands on cognitive flexibility [38], which builds on the other two EF (working memory and inhibition) and develops later [72]. The young age of TD-MA children and the more poorly developed EF of individuals with DS are therefore apparent reasons for the observed task difficulties, which are more pronounced in TMT-B, especially for the children with DS. Thus, the arguments to explain the better results of children with DS in TMT-A do not seem to apply to such a challenging task as TMT-B. Here, it can be assumed that the cognitive capacity for processing the TMT-B is limited due to the demands of higher cognitive functions (cognitive flexibility), and therefore performance is limited, especially in children with DS.

### 4.3. Relationship between the Motor and Cognitive Performance

One of the key findings of this study is the low correlation between balance and cognitive flexibility in children with DS compared to TD-CA children and the weaker correlation, in the opposite direction, between aiming and catching and cognitive flexibility in children with DS compared to TD-MA. 

Similar to the results of Schott and Klotzbier [73], the developmental trajectories of cognitive and motor performance in TD-CA children indicate comparable patterns and characteristics. Here, we observe a positive correlation between cognitive flexibility (TMT-B) and balance (MABC-2; balance percentile) of r = 0.713. The results are also consistent with van der Fels et al. [74] systematic review in TD children. The only correlations that were found in their study suggest the importance of complex motor skills and higher order cognitive skills to explain correlations between motor and cognitive skills. In contrast, the negative, low, and not significant correlation between cognitive flexibility and balance ability in children with DS (r = −0.273) suggests that either the patterns of developmental trajectories are not the same or there is greater variability in the development of both domains in children with DS [18]. A weak-to-moderate correlation was observed between cognitive functions and motor skills in children with DS [75]. Our results also align with Malak et al. [76,77] and Volman et al. [14], who found weak associations between motor and cognitive development in children with DS over six years of age. Schott and Holfelder [18] reported higher correlations than children with DS and TD-MA in this study, probably due to lower cognitive task performance variability. Both children with DS and TD-MA had significant difficulties with the TMT, especially with increasing cognitive load. The weaker and negative correlation between aiming/catching and cognitive flexibility in children with DS compared to TD-MA can possibly be explained by the fact that, in aiming and catching, children with the same mental age have not yet developed this motor skill due to a lack of practice and that the TMT-B appears to be too demanding for some of this age group.

Finally, there are some limitations and strengths of the study. Although the sample size is similar to previous studies (e.g., [78,79]), the presented results’ generalizability is limited by the sample size and the study’s cross-sectional design. Grouping children by general cognition or nonverbal developmental indices would likely be a better discriminator than receptive language. Alternative matching methods could be considered in future studies. Kover and Atwood [80] provide a brief overview of matching methods, emphasizing group matching designs used in behavioral research on cognition and language in neurodevelopmental disorders, including DS. However, the concurrent inclusion of TD-MA and TD-CA children could be mentioned as a strength. This approach makes it possible to eliminate the expected delays in motor and cognitive development (TD-MA) while having participants with more comparable biological maturation and life experience (TD-CA; [18]). Another strength worth noting is the use of the modified TMT [40,41], including a fine motor task (TMT motor speed), which allows isolating the cognitive performance of TMT-A and TMT-B by subtracting the motor speed component. However, the TMT-B may have been too demanding to evaluate EF for children with TD-MA and DS in these age ranges. In contrast to Schott and Holfelder’s study [18], the present study uses the original TMT [37], knowing that the demands may be too high for some DS and TD children of the same mental age (TD-MA) (see [81] in DS). This is true especially for tasks with high cognitive load (TMT-B) and given the evidence that reading ability (necessary for numbers and letters recognition in TMT-B) is explicitly acquired between the ages of four and six [82]. In addition, the interpretation of the test results might be problematic because, in addition to the cognitive processing speed, linguistic, executive, and attentional components are also recorded [46]. This means that differentiating the individual cognitive components required for processing appears difficult. Different components of EF play a role in processing the TMT. For example, the TMT provides information about visual search performance, information processing, fine motor skills, cognitive flexibility, and other EF [47]. It is probably the most widely used instrument for assessing task-switching ability [39,48]; Part B of the TMT is also often referred to as the “frontal lobe test” [49,50]. 

## 5. Conclusions

The results of previous studies could be confirmed separately for both domains—motor and cognitive performance. Children with DS show relative strengths in aiming and catching in the motor domain and good information-processing functioning in the cognitive domain. In higher cognitive functions (cognitive flexibility), children with DS show weak performance. Since we see a strong correlation between cognitive flexibility and balance in TD-CA children, it would be advised to improve the higher cognitive functions, especially cognitive flexibility, in children with DS in order to achieve a positive transfer effect on balance control. However, the possible influence of cognitive enhancement and transfer on balance skills needs to be investigated in randomized controlled trials.

As for the associations between the two domains, no clear picture emerges. These inconsistent results can be explained by van der Fels et al. [83] argument that different cognitive abilities are related to gross motor skills to varying degrees. This highlights a key methodological challenge of the present study. Age- and dimensionally appropriate tests that reliably measure motor and cognitive abilities are needed to capture cognitive functions, in particular EF. Similar to the age-appropriate motor skills (MABC-2) testing procedures, there should also be cognitive procedures suitable for direct comparison between TD children of different ages and children with intellectual disabilities. Our results suggest that the hypothesized relationships between motor and cognitive performance highly depend on the studied cognitive and motor skills. They imply that more specific relationships need to be investigated in future studies and that global scores or tests that can only be used to make general statements should be avoided. Therefore, it seems crucial to select a wide range of tasks for cognitive and motor domains that are as isolated as possible for future studies. While this does not always correspond to everyday tasks that combine different demands, it contributes to a better understanding of the relationships between cognitive and motor skills. Future research should consider larger sample size, different age groups, and preferably a longitudinal design to provide detailed information on the motor skills and EFs’ development trajectories to design effective interventions and optimise manual performance in individuals with DS.

## Figures and Tables

**Figure 1 children-09-00073-f001:**
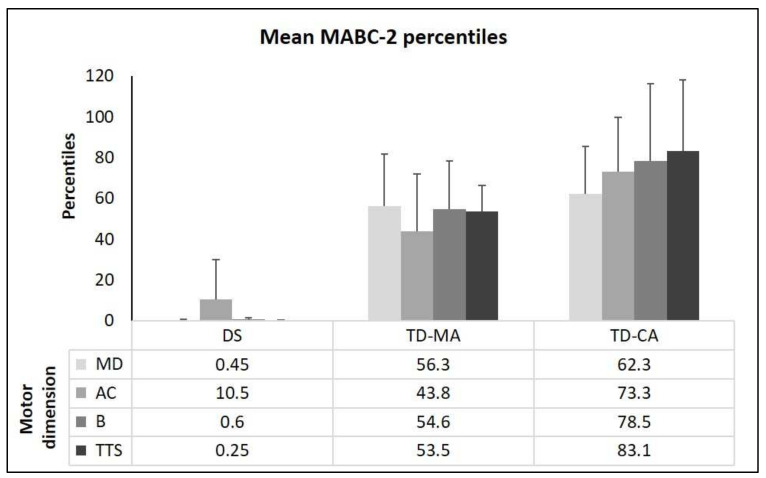
Movement Assessment Battery for Children-2 percentiles (mean + standard deviations) for the three sub-tests (MD: manual dexterity; AC: aiming and catching; and B: static and dynamic balance) and mean total test score (TTS) percentiles. Note: DS = Down Syndrome; TD-MA = Typically developing children of the same mental age; TD-CA = typically developing children of the same chronological age.

**Figure 2 children-09-00073-f002:**
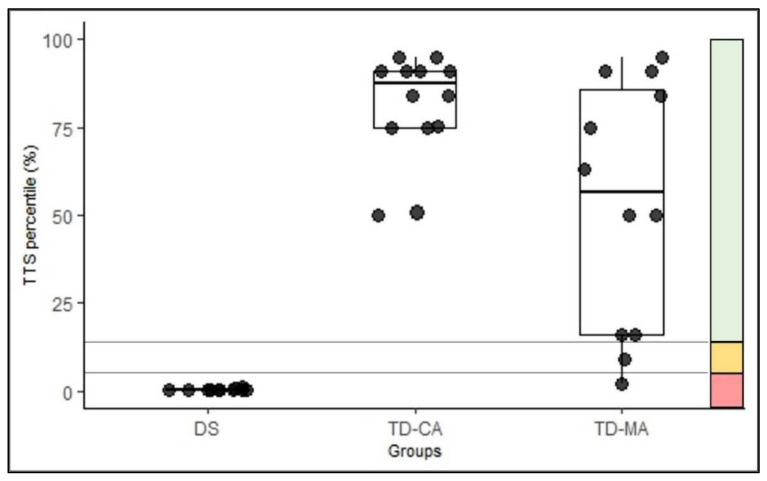
Classification of the total test score (TTS) percentiles of the groups in the traffic light system. Values ≤ 5th percentile: motor deficits (red zone); values between the 6th and 15th percentile: high-risk group (amber zone); values > 15th percentile: inconspicuous (green zone) [34]. Note: DS = Down Syndrome; TD-MA = Typically developing children of the same mental age; TD-CA = typically developing children of the same chronological age.

**Figure 3 children-09-00073-f003:**
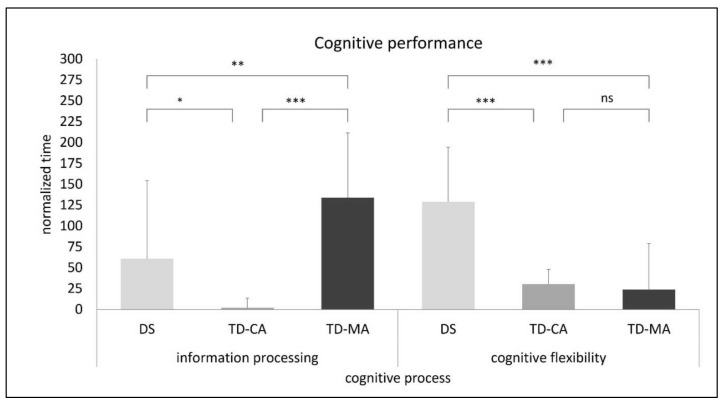
Performance in information processing and cognitive flexibility based on the normalized times in the Trail-Making Test (ns: not significant; *** *p* < 0.001; ** *p* < 0.01; * *p* < 0.05). Note: DS = Down Syndrome; TD-CA = Typically developing children of the same chronological age; TD-MA = Typically developing children of the same mental age.

**Table 1 children-09-00073-t001:** Sampling characteristics of children with and without DS (adjusted chronological age (CA); adjusted mental age (MA)), including mean values (standard deviation) and receptive vocabulary scores of the Peabody Picture Vocabulary Test, 4th edition (PPVT-IV).

	DS	TD-CA	TD-MA	Statistical Analyses
	(*n* = 12)	(*n* = 12)	(*n* = 12)	
Age (years)	10.5 ± 10.08 ^§^	10.5 ± 10.07	5.98 ± 1.21 ^#^	*F*(2,33) = 65.8, *p* < 0.001, η^2^_p_ = 0.799
Sex (% male)	500.0	500.0	500.0	CHI^2^(2) = 00.00, *p* = 10.00
Weight (kg)	32.1 ± 7.76 ^§,#^	40.1 ± 7.67	19.4 ± 50.02 ^#^	*F*(2,33) = 27.1, *p* < 0.001, η^2^_p_ = 0.621
Height (cm)	133 ± 80.05 ^§,#^	150 ± 5.83	115 ± 13.8 ^#^	*F*(2,33) = 37.9, *p* < 0.001, η^2^_p_ = 0.697
BMI (kg/m^2^)	180.0 ± 2.76 ^§^	17.8 ± 2.73	14.6 ± 1.62 ^#^	*F*(2,33) = 7.27, *p* = 0.002, η^2^_p_ = 0.306
Medication (n)	1.42 ± 0.79 ^§,#^	00.00 ± 00.00	00.08 ± 0.29	*F*(2,33) = 31.9, *p* < 0.001, η^2^_p_ = 0.659
Finger length (cm)	5.48 ± 0.34 ^#^	7.13 ± 0.53	5.38 ± 0.70 ^#^	*F*(2,33) = 39.2, *p* < 0.001, η^2^_p_ = 0.704
Palm length (cm)	80.07 ± 0.93^#^	9.54 ± 0.50	7.46 ± 0.84^#^	*F*(2,33) = 22.8, *p* < 0.001, η^2^_p_ = 0.580
Palm-to-finger length ratio	0.69 ± 00.07	0.75 ± 00.05	0.72 ± 00.07	*F*(2,33) = 2.80, *p* = 0.075, η^2^_p_ = 0.145
PPVT-IV raw value Receptive vocabulary Score	96.6 ± 19.7 ^#^ 66.6 ± 2.13	172 ± 190.0 980.0 ± 12.2	105 ± 28.4 ^#^ 920.0 ± 10.4	*F*(2,33) = 38.9, *p* < 0.001, η^2^_p_ = 0.702 *F*(2,33) = 38.1, *p* < 0.001, η^2^_p_ = 0.698
Sports participation (min/week)	138 ± 45.1 ^§^	158 ± 71.4	62.5 ± 71.4 ^#^	*F*(2,33) = 7.38, *p* = 0.002, η^2^_p_ = 0.309

Note. BMI = Body Mass Index; ^#^ Significant difference to CA-adjusted group (*p* < 0.05); ^§^ Significant difference to MA-adjusted group (*p* < 0.05).

**Table 2 children-09-00073-t002:** Partial correlations (*r*) across cognitive (Trail-Making Test) and motor (Movement Assessment Battery for Children-2) indices for the sample of TD-MA (*n* = 12), TD-CA (*n* = 12) and children with DS (*n* = 12) controlled for sports participation and sex.

	MD Percentile	AC Percentile	B Percentile	TTS Percentile
	DS			
	*r*	*r*	*r*	*r*
TMT-M	0.177	−0.179	0.032	−0.206
TMT-A	−0.038	0.074	−0.190	−0.215
TMT-B	−0.232	0.367	−0.273	−0.189
	TD-MA			
TMT-M	0.156	0.004	0.318	0.158
TMT-A	0.078	−0.184	−0.204	−0.170
TMT-B	−0.317	−0.401	−0.520 ^T^	−0.456 ^T^
	TD−CA			
TMT-M	−0.280	0.418	0.032	0.189
TMT-A	−0.385	−0.407	0.345	−0.570 *
TMT-B	−0.335	−0.089	0.713 *	−0.316

Note. MD = manual dexterity; AC = aiming and catching; B = balance; TTS = Total Test Score; TMT-M = Trail-Making Test, single motor task; TMT-A = Trail-Making Test, information processing; TMT-B = Trail-Making Test, cognitive flexibility; DS = Down Syndrome; TD-MA = Typically developing children of the same mental age; TD-CA = Typically developing children of the same chronological age; *r* = partial correlation; * *p* < 0.05, ^T^ <0.10; a log (Base 10) transformation is be applied to each participant’s velocity score to create a more normal distribution of scores.

**Table 3 children-09-00073-t003:** Differences in magnitude of correlations (Fisher’s z) across cognitive (Trail-Making Test) and motor (Movement Assessment Battery for Children-2) indices for the sample of TD-MA (*n* = 12), TD-CA (*n* = 12) and children with DS (*n* = 12) controlled for sports participation and sex.

	DS vs. TD-MA			
	MD Percentile	AC Percentile	B Percentile	TTS Percentile
TMT-M	z = 0.046; *p* = 0.481	z = −0.394; *p* = 0.346	z = −0.631; *p* = 0.264	z = −0.781; *p* = 0.217
TMT-A	z = −0.246; *p* = 0.402	z = 0.554; *p* = 0.289	z = 0.031; *p* = 0.487	z = −0.099; *p* = 0.460
TMT-B	z = 0.354; *p* = 0.361	z = 1.72; *p* = 0.042	z = 0.628; *p* = 0.264	z = 0.638; *p* = 0.261
	DS vs. TD−CA			
TMT-M	z = 0.99; *p* =.161	z = −1.33; *p* = 0.091	z = 0.0; *p* = 0.5	z = −0.849; *p* = 0.197
TMT-A	z = 0.78; *p* = 0.217	z = 1.16; *p* = 0.123	z = −1.171; *p* = 0.120	z = 0.91; *p* = 0.181
TMT-B	z = 0.238; *p* = 0.405	z = 10.01; *p* = 0.156	z = −2.489; *p* = 0.006	z = 0.288; *p* = 0.386
	TD−MA vs. TD−CA			
TMT-M	z = 0.944; *p* = 0.172	z = −0.936; *p* = 0.174	z = 0.631; *p* = 0.264	z = −0.068; *p* = 0.472
TMT-A	z = 10.03; *p* = 0.152	z = 0.522; *p* = 0.301	z = −1.20; *p* = 0.114	z = 10.01; *p* = 0.156
TMT-B	z = 0.043; *p* = 0.482	z = −0.712; *p* = 0.238	z = −3.12; *p* < 0.001	z = −0.35; *p* = 0.363

Note. MD = manual dexterity; AC = aiming and catching; B = balance; TTS = Total Test Score; TMT-M = Trail-Making Test, single motor task; TMT-A = Trail-Making Test, information processing; TMT-B = Trail-Making Test, cognitive flexibility; DS = Down Syndrome; TD-MA = Typically developing children of the same mental age; TD-CA = Typically developing children of the same chronological age; by convention, values greater than |1.64| are considered significant in a 1-tailed test.

## Data Availability

All relevant data are within the study and raw data are available on request.

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
