# Peer review of "Associations of Motor Performance and Executive Functions: Comparing Children with Down Syndrome to Chronological and Mental Age-Matched Controls"

_children, 2022, doi:10.3390/children9010073_

Round 1

Reviewer 1 Report

The purpose of this study was to examine motor skills and executive function, and their relation, in a sample of participants with Down syndrome. Comparison groups with typical development, matched on mental age and chronological age, outperformed the group with Down syndrome on most measures. However, when examining group differences in the magnitude of correlations between motor skill and executive function, there were only two statistically significant findings relevant to the group with Down syndrome (weaker correlation between balance and cognitive flexibility vs. TD-CA and a weaker correlation, also in the opposite direction, between aiming and catching and cognitive flexibility vs. TD-MA). Motor skills are important for learning and social interactions. Thus, this study addresses an important research question. However, there are several weaknesses that need to be addressed before it is ready for publication.

Introduction

  1. The introduction needs greater development and to be re-worked for clarity and flow. Edit the introduction to focus on one main point per paragraph (e.g., the second paragraph makes multiple different “key” points).
  2. The introduction would also be strengthened by an overview of different motor skills and their development, both normal motor skill development and what we know about these skills in individuals with DS.
  3. What is known about executive function in DS? There is a decent body of research examining profiles of strengths and weaknesses across different executive function skills in this population, and there is more to say than just that they have “reduced” ability in EF (e.g., the authors citation of Tungate and Conners, 2021 is a great place to start).
  4. “Intellectual disability” is the appropriate term, not “intellectually challenged”. Also, the authors should edit to use person-first language throughout.

Methods

  1. Participants – what was the justification for the 8-12-year-old age range of participants with DS? This should have been justified, at least partially, in the introduction. Also, explain how each of the inclusion criteria was checked/verified.
  2. Participants – much more information is needed about the matching procedures. Please clarify and edit this point, “Based on PPVT-IV results, TD children were assigned to the control groups,”, as the CA group was not matched in this way. Perhaps more concerning is that while the authors note that the PPVT-IV was used to “match” the participants on MA and that CA was used to “match” participants on CA, there is little information regarding how this matching was determined. The authors should explain their matching procedures and demonstrate that the groups are truly matched, not just that the groups are not significantly different on the matching variable (see Kover & Atwood, for guidance). I suggest adding a “matching” section to explicitly note how matching occurred.
    1. Kover, S. T., & Atwood, A. K. (2013). Establishing equivalence: Methodological progress in group-matching design and analysis. American Journal on Intellectual and Developmental Disabilities118(1), 3-15.
  3. Participants – if possible, please report race and ethnicity data for participants.
  4. Measures – please report reliability and validity information of all measures.
  5. Measures – when describing the PPVT-IV, I don’t believe Loveall et al. (2016) established that the PPVT can be used as a measure of verbal intelligence. Perhaps they used the PPVT-IV, but did they actually test its validity as a measure of verbal intelligence?
  6. Measures – the authors use the Trail-Making-Task as a measure of executive function, defined in their methods as information processing and cognitive flexibility. However, their definition of executive function in the introduction (cognitive flexibility, inhibition, and working memory) did not include information processing. Please make this link explicit.
  7. Procedures – what was the order of tasks? If not randomized or counter-balanced, please justify.
  8. Procedures – did participants receive an incentive to participate? Also, it is noted that parents signed a consent form. Did children also go through a verbal assent procedure?
  9. Data Analysis – the study includes a very small sample, yet the authors are conducting fairly sophisticated statistical analysis. Please provide a power analysis and/or power report to indicate that the study is appropriately powered to detect significant differences between groups. Alternatively, the authors could recruit and test a larger sample.
  10. Data analysis – was there any missing data? And if so, how was it handled? Similarly, were any participants unable to complete testing for any reason?
  11. Data analysis – were all assumptions tested for and met for all analyses (e.g., MANOVA, ANOVA)?

Results

  1. the MANOVA and ANOVA results are a little difficult to follow in the text, mostly because there are so many results. I think these could be easier to track in a table format.

Discussion

  1. I think the discussion could be strengthened by talking about patterns of relative strength and areas of weakness across motor and executive function skills for the group with Down syndrome.
  2. What are some implications or clinical applications of the findings?

Minor Edits

  1. Abstract – in the first sentence the authors say that children with DS exhibit poorer motor and cognitive performance than TD children. In the next sentence, they say only a few studies have addressed this in relation to CA or MA matches with TD. What comparison group (with TD) does the first sentence refer to then?
  2. Abstract – the “s” in Down syndrome should not be capitalized.
  3. To improve readability, I suggest reducing the number of acronyms used throughout the manuscript. I also suggest reducing the length of some paragraphs (e.g., by dividing into smaller paragraphs).

Author Response

Reviewer 1

The purpose of this study was to examine motor skills and executive function, and their relation, in a sample of participants with Down syndrome. Comparison groups with typical development, matched on mental age and chronological age, outperformed the group with Down syndrome on most measures. However, when examining group differences in the magnitude of correlations between motor skill and executive function, there were only two statistically significant findings relevant to the group with Down syndrome (weaker correlation between balance and cognitive flexibility vs. TD-CA and a weaker correlation, also in the opposite direction, between aiming and catching and cognitive flexibility vs. TD-MA). Motor skills are important for learning and social interactions. Thus, this study addresses an important research question. However, there are several weaknesses that need to be addressed before it is ready for publication.

Dear Reviewer,

First of all, thank you for the constructive feedback on our article: "Associations of Motor Performance and Executive Functions: Comparing Children with Down Syndrome to Chronological and Mental Age-Matched Controls".

We have addressed all you comments in the datail and think that this hassignificantly enhanced the manuscript.

Especially with regard to the structuring of the paragraphs in the introduction, which improves the flow of reading, as well as through the special emphasis on the central results regarding the correlations in the discussion.

We have additionally adjusted the figures. We think especially figure 2 is now more informative due to the individual points (jitter plot) and the displayed boxplots. Figures 1 and 3 were primarily enhanced in colour.

We have highlighted some implications that can be drawn from our findings.

Below you can see our responses to your comments. In the document we have highlighted our changes using Word's correction mode.

We thank you again for your feedback

Introduction

The introduction needs greater development and to be re-worked for clarity and flow. Edit the introduction to focus on one main point per paragraph (e.g., the second paragraph makes multiple different “key” points).

  • We have structured the paragraphs more clearly. In the first paragraph, we discuss the genetic neurodevelopmental disorder DS and describe the most obvious characteristics of people with DS, before moving on to the second paragraph, which deals with the affected cognitive areas and briefly describes the construct of EF. In a third paragraph, impairments in the motor domain are described. In a fourth section, we discuss the cross-domain relationship between motor and cognitive performance in children with intellectual disabilities and review studies that have investigated this relationship. In a fifth paragraph, we try to briefly describe the methodological gaps, according to which the few studies with DS use heterogeneous methods to investigate motor and cognitive performance, which makes it difficult to generalise the results. We then go into more detail on the study by Schott and Holfelder (2015), which uses a comparable study design. Then, in a sixth paragraph, we describe what we do/adapt differently compared to previous studies (especially compared to Schott and Holfelder (2015)). More specifically, we want to use a group with the same mental age and justify this with the possibility of using the explanatory approaches of "Conventional Difference Hypothesis" postulated by Milgram (2017) and "Unconventional Differential Hypothesis" postulated by Kohlberg (1968).

In a final paragraph, we then come to our objectives and the expectations based on the current state of research and the mentioned developmental hypotheses.

The introduction would also be strengthened by an overview of different motor skills and their development, both normal motor skill development and what we know about these skills in individuals with DS.

  • We have restructured the beginning of the introduction by specifically addressing cognitive and motor limitations.
  • We have added the following sentence in the third paragraph: „Uncoordinated, slower, more variable, and hesitant movements, along with a poor ability to respond to environmental changes, are characteristics of motor skill perfor-mance in children with DS [11]. Additionally, motor abilities like coordination, balance and strength [12] may not be as developed as the ones peers without DS would show. Using various approaches, further studies showed that children with DS between the ages of 6 and 16 years performed worse on fundamental movement skills than their typically developing peers [13-15]. Volman et al. [14] observed that children with DS scored poorly on manual dexterity, followed by balance and then ball skills with high interindividual variability.“ (page 2 line 47-55)

What is known about executive function in DS? There is a decent body of research examining profiles of strengths and weaknesses across different executive function skills in this population, and there is more to say than just that they have “reduced” ability in EF (e.g., the authors citation of Tungate and Conners, 2021 is a great place to start).

  • We have added the following paragraph to highlight the cognitive characteristics of individuals with DS: Regarding cognition, the main areas affected are language skills, processing speed, attention processes, visuo-spatial abilities and specifically a reduced ability in executive functions (EF) [5, 6]. The construct of EF has received considerable attention over the last three decades. Despite, or perhaps because of, the many publications, there is no gen-erally accepted definition of EF [7]. Frequently, EF are divided into three domains: cog-nitive flexibility, inhibition ability, and working memory, and encompasses an extensive set of higher-order operations that organise and regulate goal-directed behaviour within the prefrontal cortex (PFC) [8]. In their meta-analysis, Tungate and Conners [5] were able to show that there is a clinically significant overall weakness in EF for individuals DS with a relative strength in inhibition ability relative to TD children with the same mental age. (page 2 line 34-43)
  • We also describe more specifically what was found in the study by Tungate and Conners (2021).

“Intellectual disability” is the appropriate term, not “intellectually challenged”. Also, the authors should edit to use person-first language throughout.

  • Thank you for pointing this out. We have changed "intellectually challenged" to "intellectually disabled" (page 2 line 67) and use person-first language throughout the manuscript. Instead of „DS children“ we now write „children with DS“.

Methods

Participants – what was the justification for the 8-12-year-old age range of participants with DS? This should have been justified, at least partially, in the introduction. Also, explain how each of the inclusion criteria was checked/verified.

  • The aim was to compare the results with those of Holfelder and Schott [18] with a similar study design. A second reason was the fact that children with DS are only able to perform the TMT in this age range. In addition, the comparison group of children with the same mental age is still able to perform the TMT.
  • We added the following information to the description of the inclusion criteria: “ (to compare the results by Schott and Holfelder [18] and due to the fact that children with DS are only able to perform the TMT at this age range)". (page 3 line 126-128)
  • We have also added the following sentence to describe how the inclusion criteria were verified: „All the inclusion criteria mentioned were verified by asking the parents and the edu-cators.“ (page 3 line 134)

Participants – much more information is needed about the matching procedures. Please clarify and edit this point, “Based on PPVT-IV results, TD children were assigned to the control groups,”, as the CA group was not matched in this way. Perhaps more concerning is that while the authors note that the PPVT-IV was used to “match” the participants on MA and that CA was used to “match” participants on CA, there is little information regarding how this matching was determined. The authors should explain their matching procedures and demonstrate that the groups are truly matched, not just that the groups are not significantly different on the matching variable (see Kover & Atwood, for guidance). I suggest adding a “matching” section to explicitly note how matching occurred.

Kover, S. T., & Atwood, A. K. (2013). Establishing equivalence: Methodological progress in group-matching design and analysis. American Journal on Intellectual and Developmental Disabilities118(1), 3-15.

  • Thank you for the advice and reference regarding the matching of the groups. We have included a chapter: "2.1.1.Matching Procedure“ (page 3 line 135) àA TD child was included in the TD-MA control group if their raw score in the PPVT-IV was less than four standard deviation points (within the 4 SD range of DS children) away from the corresponding mean score of children with DS. A TD child was included in the TD-CA group if their CA was within the 4-month range of the DS children.“ (page 3 line 136-140)
  • We have also included the following sentence in the limitations and refer to the article by Kover, and Atwood (2013) who propose other matching indices: „Alternative matching methods could be considered in future studies. Kover and At-wood [80] provide a brief overview of matching methods, emphasising group matching designs used in behavioural research on cognition and language in neurodevelopmental disorders, including DS.“ (page 12 line 447-450)

Participants – if possible, please report race and ethnicity data for participants.

  • We added the information „Caucasian“ (page 3 line 118). We did not asked the parents for further information about race and ethnicity. It is a sensible topic in our country because oft he German history. There is discussion about completely removing the concept of race from the consitution law in Germany.

Measures – please report reliability and validity information of all measures.

  • We have added information on the psychometric properties for the TMT and the MABC-2:
  • MABC: „The test reliability after two weeks is r = .97 (N = 138; [34]). According to Blank et al. [36] the MABC-2 show good to excellent interrater reliability, good to excellent test–retest reliability and fair to good validity.“ (page 4 line 161-163)
  • TMT: „A value of r = .94 (TMT-A) to .90 (TMT-B) were calculated for the test reliability of the TMT [43]. For background, psychometric properties, administration procedures and interpretive guidelines of the TMT we refer to [44] and [45].“ (page 4 line 185-187)

Measures – when describing the PPVT-IV, I don’t believe Loveall et al. (2016) established that the PPVT can be used as a measure of verbal intelligence. Perhaps they used the PPVT-IV, but did they actually test its validity as a measure of verbal intelligence?

  • That's absolutely correct. Very attentive. Thank you for the feedback. We have changed the sentence and now write: „It can be utilised to measure the receptive processing of vocabulary in individuals with ID [51].“ (page 5 line 209-210)
  • We also refer here to the study by Krasileva et al (2017) who use the PVT score as a proxy for IQ: „Krasileva and colleagues [52] used the PPVT-4 scores as proxy for IQ in studies of autism spectrum disorder.“ (page 5 line 210-212)

Measures – the authors use the Trail-Making-Task as a measure of executive function, defined in their methods as information processing and cognitive flexibility. However, their definition of executive function in the introduction (cognitive flexibility, inhibition, and working memory) did not include information processing. Please make this link explicit.

  • We agree that we have adopted this simple division into three parts of the EF. We already write: „Despite, or perhaps because of, the many publications, there is no generally accepted definition of EF [7].“ (page 1 line 37-38)
  • Information processing is not EF, but correlative studies suggest that it is related to EF. We operationalise EF primarily through TMT-B. We are interested in measuring information processing speed with TMT A and EF with TMT B.*
  • The interpretation of the test results is not that simple, since in addition to the cognitive processing speed, linguistic, executive and attentional components are recorded (Salthouse, 2011). This means that it is difficult to differentiate between the individual cognitive components that are required to complete the test. Various components of EF play a role in the processing of the TMT. For this reason, we have added the following sentence to the description of the TMT: „In addition to cognitive processing speed, also linguistic, executive and attentional components are assessed [46]. Thus, it is difficult to distinguish between the different cognitive components that are required to complete the TMT. Various components of EF play a role in processing the TMT. For example, the TMT provides information on per-formance in visual search, information processing, fine motor skills, cognitive flexibility and other EF [47]. It is probably the most widely used instrument for assessing task switching ability [39, 48] and Part B of the TMT is also often referred to as the "frontal lobe test" [49, 50], which is strongly associated with EF.“ (page 4 line 188-195)

Procedures – what was the order of tasks? If not randomized or counter-balanced, please justify.

  • This is an important methodological information. We have added the following sentence: „To avoid cognitive and physical fatigue from collecting data through the MABC-2 and to avoid loss of attention that could influence performance, the cognitively demanding TMT was conducted for all participants prior to the MABC-2.“ (page 5 line 228-230)

Procedures – did participants receive an incentive to participate? Also, it is noted that parents signed a consent form. Did children also go through a verbal assent procedure?

  • Thank you for your comments. We have included them in our methods. We have adjusted the sentence to make clear that the children have been asked for their consent: " The children were asked for their consent and their willingness to participate in the study and the participant's legal guardian/next of kin provided written informed con-sent to participate in this study.“ (page 5 line 243-247)
  • We also mentioned that the children or legal guardians did not receive any financial compensation for participating in the study: „The participants or the legal guardians of the children did not receive any financial compensation or incentive for taking part in the study.“ (page 6 line 245-247)

Data Analysis – the study includes a very small sample, yet the authors are conducting fairly sophisticated statistical analysis. Please provide a power analysis and/or power report to indicate that the study is appropriately powered to detect significant differences between groups. Alternatively, the authors could recruit and test a larger sample.

  • We are aware of the limitation that the statistical power would benefit from more study participants. For this reason, we have already mentioned the limitations and referred to comparable studies with similar sample sizes.: „Although the sample size is similar to previous studies (e.g., [78, 79]), the presented results' generalizability is limited by the sample size and the study's cross-sectional design.“ (page 12 line 443-445)
  • We also mentioned in the Conclusions that in future studies a larger sample will lead to more meaningful and stable effects: „Future research should consider larger sample size, different age groups, […].“ (page 13 line 597-500)
  • Unfortunately, we were not able to recruit further study participants with DS, as this is a specific sample population. This is also the reason why it is difficult to collect further data, which is additionally complicated by the circumstances (current Corona situation). We also believe that the further collection of data would reduce the comparability within the study design.
  • From a methodological point of view, based on the calculated effect sizes in our results, it can be concluded that the sample size was appropriate to justify the statistical methods.

 Data analysis – was there any missing data? And if so, how was it handled? Similarly, were any participants unable to complete testing for any reason?

  • We have no missing values in our data set. This is indicated by the degrees of freedom of the ANOVAs. We have added the sentence: "There were no missing data" in the description of the statistics.(page 6 line 256)

All participants were able to complete the data collection from the beginning to the end without interrupting or cancelling.

Data analysis – were all assumptions tested for and met for all analyses (e.g., MANOVA, ANOVA)?

  • As far as we have understood your question, we can answer this with yes. This applies to the group differences as well as to the calculation of the correlations.

Results

The MANOVA and ANOVA results are a little difficult to follow in the text, mostly because there are so many results. I think these could be easier to track in a table format.

  • We also see that the results are very extensive. However, we have presented the key results in a chart. For a clear presentation of the extensive results, we have decided on a balanced presentation in the form of tables, figures and text. A focus on table-based presentations would disrupt this relationship or even lead to redundancies. For this reason, we would prefer and retain the current representation of the results.

Discussion

I think the discussion could be strengthened by talking about patterns of relative strength and areas of weakness across motor and executive function skills for the group with Down syndrome.

  • Thank you for pointing this out. When proofreading the discussion (especially in the chapter: Relationship between the motor and cognitive performance), we also noticed that the key results are formulated too vaguely. We have now done this in the discussion chapter: „One of the key findings of this study is the low correlation between balance and cognitive flexibility in children with DS compared to TD-CA children and the weaker correlation, in the opposite direction, between aiming and catching and cognitive flex-ibility in children with DS compared to TD-MA.“ (page 11 line 418-421)

What are some implications or clinical applications of the findings?

  • Children with DS have relative strengths in information processing and aiming and catchung. They show weaknesses in higher cognitive functions and balance performance. A possible implication would be to improve higher cognitive functions in order to obtain a positive transfer to balance performance. In our study we do not observe a correlation between these domains in children with DS. This low correlation is probably due to the consistently low performance of children with DS in TMT-B (cognitive flexibility). We write:“ Children with DS show relative strengths in aiming and catching in the motor domain and good information processing functioning in the cognitive domain. In higher cogni-tive functions (cognitive flexibility), children with DS show weak performance. Since we see a strong correlation between cognitive flexibility and balance in TD-CA children, it would be advised to improve the higher cognitive functions, especially cognitive flexibility, in children with DS in order to achieve a positive transfer effect on balance control. However, the possible influence of cognitive enhancement and transfer on balance skills needs to be investigated in randomized controlled trials.“ (page 12 line 474-481)

Minor Edits

Abstract – in the first sentence the authors say that children with DS exhibit poorer motor and cognitive performance than TD children. In the next sentence, they say only a few studies have addressed this in relation to CA or MA matches with TD. What comparison group (with TD) does the first sentence refer to then?

  • The point is that few studies have an MA and a CA control group. We have adapted the sentence and write. „Although there is a relationship between these two developmental domains, only few studies have addressed this association in children with DS compared to groups of the same chronological- (CA) or mental-age (MA) within one study“ (abstract line 14)

Abstract – the “s” in Down syndrome should not be capitalized.

  • Thank you. The "s" in Down syndrome is written in lower case (abstract line 11)

To improve readability, I suggest reducing the number of acronyms used throughout the manuscript. I also suggest reducing the length of some paragraphs (e.g., by dividing into smaller paragraphs).

  • Especially in the introduction, we have adjusted the length of some paragraphs. We have restructured this paragraph, which significantly improves readability.
  • We have written out the abbreviations of the MABC categories MD = manual dexterity, AC = aiming and catching, B = balance.
  • We have abbreviated the terms of the TMT (TMT-M, TMT-A and TMT-B), as this term appears frequently and would increase the length of the paper considerably. The same applies to the abbreviations of the groups (DS, TD-MA and TD-CA).

Reviewer 2 Report

I don´t understand why Peabody has been used in mental age. Peabody test is assess passive vocabulary. In no case, is a measure about mental age. The authors must explain this decision.

There is no information about medium ages in every group. The results couldn´t be correct if this part isn´t explained with more detail.

Conclussion could be improved with more bibliography abour relationshio between motor skills and executive functions in intellectual disability.

The simple is very small and it´s difficult to generalize the results.

Author Response

Reviewer 2

Dear Reviewer,

First of all, thank you for the constructive feedback on our article: "Associations of Motor Performance and Executive Functions: Comparing Children with Down Syndrome to Chronological and Mental Age-Matched Controls".

We have addressed all you comments in the datail and think that this has significantly enhanced the manuscript.

Especially with regard to the structuring of the paragraphs in the introduction, which improves the flow of reading, as well as through the special emphasis on the central results regarding the correlations in the discussion. We have additionally adjusted the figures. We think especially figure 2 is now more informative due to the individual points (jitter plot) and the displayed boxplots. Figures 1 and 3 were primarily enhanced in colour.

We used the PPVT score as a proxy for IQ and therefore for mental age. Also, we mentioned in the limitations that grouping based on general intelligence is probably a better discriminator.

We have highlighted some implications that can be drawn from our findings. Was ist das?

Below you can see our responses to your comments. In the document we have highlighted our changes using Word's correction mode.

We thank you again for your feedback

I don´t understand why Peabody has been used in mental age. Peabody test is assess passive vocabulary. In no case, is a measure about mental age. The authors must explain this decision.

  • Mental age is a concept related to intelligence. It looks at how a specific individual, at a specific age, performs intellectually, compared to average intellectual performance for that individual's actual chronological age (i.e. time elapsed since birth).
  • Children with DS, who usually have a mild to moderate intelligence impairment, are also delayed in their language development. Both cognitive and language development are significantly slower than in typically developing children. The gap with chronological age thus continues to increase (Patterson, Rapsey, & Glue, 2013*). According to Darmer (2018), the age of lexical development in childhood is approximately half of the chronological age.
  • We used the PPVT score as a proxy for IQ and therefore for mental age.

*Patterson, T., Rapsey, C. M., & Glue, P. (2013). Systematic review of cognitive development across childhood in Down syndrome: implications for treatment interventions. Journal of Intellectual Disability Research, 57 (4), 306-318.

*Darmer, A. (2018). Die Entwicklung des produktiven Wortschatzes von Kindern und Jugendlichen mit Down Syndrom – Ein systematisches Review. Logos, 26 (1), 4-14.

Krasileva et al 2017 investigated the validity of the Peabody Picture Vocabulary Test-4th Edition (PPVT-4; Dunn and Dunn 2007) as a low-cost, less time-consuming alternative to traditional cognitive batteries, to obtain an approximate value for verbal IQ (VIQ)

*Dunn, L. M., & Dunn, L. M. (1981). Peabody picture vocabulary test—Revised. Circle Pines, MN: American Guidance Service.

*Dunn, L. M., & Dunn, L. M. (1997). Peabody picture vocabulary test, third edition. Circle Pines, MN: AGS.

Also, we mentioned in the limitations that grouping based on general intelligence is probably a better discriminator: „Grouping children by general cognition or nonverbal developmental indices would likely be a better discriminator than receptive language. Alternative matching methods could be considered in future studies. Kover and Atwood [80] provide a brief overview of matching methods, emphasising group matching designs used in behavioural re-search on cognition and language in neurodevelopmental disorders, including DS.“ (page 12 line 445-450)

There is no information about medium ages in every group. The results couldn´t be correct if this part isn´t explained with more detail.

  • We have reported the mean values and standard deviations of the age for the groups in the sample characteristics (Table 1). Additional information is now in the body text: „The mean age of children with DS is 10.5 ± 1.08, TD-CA children 10.5 ± 1.07 and TD-MA children 5.98 ± 1.21. (page 6 line 274-275; and table 1)

Conclussion could be improved with more bibliography abour relationshio between motor skills and executive functions in intellectual disability.

  • Thank you for the comment. Upon reading the discussion on the correlations again, we also noticed that we only vaguely formulate the central findings.
  • We have adjusted the structure of the discussion on the relationships and we propose to conduct further studies to show whether exercising higher cognitive functions has a transfer effect on balance ability in children. This serves as a possible recommendation and future direction based on our findings.

The simple is very small and it´s difficult to generalize the results.

  • We are aware of the limitation that the statistical power would benefit from more study participants. For this reason, we have already mentioned the limitations and referred to comparable studies with similar sample sizes.: „Although the sample size is similar to previous studies (e.g., [78, 79]), the presented results' generalizability is limited by the sample size and the study's cross-sectional design.“ (page 12 line 443-445)
  • We also mentioned in the Conclusions that in future studies a larger sample will lead to more meaningful and stable effects: „Future research should consider larger sample size, different age groups, […].“ (page 13 line 497-500)
  • Unfortunately, we were not able to recruit further study participants with DS, as this is a specific sample population. This is also the reason why it is difficult to collect further data, which is additionally complicated by the circumstances (current Corona situation). We also believe that the further collection of data would reduce the comparability within the study design.
  • From a methodological point of view, based on the calculated effect sizes in our results, it can be concluded that the sample size was appropriate to justify the statistical methods.

Round 2

Reviewer 2 Report

The article has improved clearly. Congrats!